# Shaped Microwave Field in a Three-Level Closed Loop Dense Atomic System

**DOI:** 10.3390/molecules28052096

**Published:** 2023-02-23

**Authors:** Nadia Boutabba, Hazrat Ali

**Affiliations:** 1Institute of Applied Technology, Fatima College of Health Sciences, Abu Dhabi P.O. Box 3798, United Arab Emirates; 2Department of Physics, Abbottabad University of Science and Technology, Havellian P.O. Box 22500, Pakistan

**Keywords:** three-level atomic systems, density matrix, microwave field, pulse shaping

## Abstract

In this work, we investigate the atomic properties of a three-level system under the effect of a shaped microwave field. The system is simultaneously driven by a powerful laser pulse and a weak constant probe that drives the ground state to an upper level. Meanwhile, an external microwave field drives the upper state to the middle transition with shaped waveforms. Hence, two situations are considered: one in which the atomic system is controlled by a strong laser pump and a classical constant microwave field, and another in which both the microwave and pump laser fields are shaped. Finally, for sake of comparison, we investigate the tanh-hyperbolic, the Gaussian and the power of the exponential microwave form in the system. Our results reveal that shaping the external microwave field has a significant impact on the absorption and dispersion coefficient dynamics. In comparison with the classical scenario, where usually the strong pump laser is considered to have a major role in controlling the absorption spectrum, we show that shaping the microwave field leads to distinct results.

## 1. Introduction

The use of laser light to control atoms and molecules has extensively been investigated in quantum optics [1,2,3,4,5,6] and molecular physics [6]. For instance, coherent control of light (OCT) has made significant contributions to our understanding of the manipulation of photo-chemical reactions [7,8]. In the experimental field of optimal control of light, a short laser pulse can drive optical or infrared transitions with the aim to optimize a specific yield of a photo-chemical reaction [9]. This can be accomplished by creating interference between light-induced pathways or by focusing wave packets in a certain direction [8]. Additionally, other impressive fields of research based on the (OCT) emerged, such as quantum computing, quantum information and quantum circuits [10,11].

To obtain effective coherent light control, one key strategy is to shape the pulse, which attempts to create ultra-fast and programmable signals. Indeed, chemically selective non-linear spectroscopy has been enabled through the use of pulse shaping, single laser beams, and coherent anti-Stokes Raman scattering (CARS) [12]. The chirp frequency was used to shape pulses and thus to analyze the population of higher energy levels’ dependence on different light frequencies. In fact, numerous shaped pulses were used to control a multi-level atom’s negative refractive index, the atomic population inversion, coherence, and absorption–dispersion spectra [13,14,15]. Those pulses attempt to control the atomic medium through the strong laser beam. This leads to the manipulation of the atomic coherence, which is induced by interference pathways and hence, allows the Raman transition of the atom to be equal to the frequency difference between the strong and the weak probe fields. As a result, a transparency window in the atomic spectra, known as the electromagnetically induced transparency, is formed. In this context, the dynamical behavior of both the dispersion and absorption depends on the spontaneously generated coherence (SGC) and the phase mismatch between the applied laser beams, which implies that the medium can be used, for example, as an optical switch [16]. Moreover, the manipulation of the coherence is a key tool to slow the light by adjusting the group velocity [17], whereas the optical Kerr-effect (OKE) [5] is based on a third-order non-linear process caused by the third-order term of polarization expansion (related to the third-order term of coherence) and can be controlled by coherence manipulation.

From another side, microwave signal synthesis is of tremendous interest in a variety of research domains, including coherent light control, wireless networks, wireless software, antennas, radar, and optical sensors. These signals can be generated experimentally using an intensity phase modulator in combination with a reference source [18] based on the technology of single light sources [18]. In addition, arbitrary waveforms at high frequencies can be generated using the pulse-shaping technology with no requirement of a reference light source [19].

Furthermore, the induced change in the probe absorption between the levels that are not driven by the strong control laser field is the basis of the EIT in multi-level systems. The reason for this is that a low-frequency field increases the generated atomic coherence due to the interference between the different atomic states [20]. Manipulating the amplitude and the phase of a microwave signal in EIT efficiently enables pertinent phenomena in the spontaneous emission spectra, for instance, spectral line suppression, side band narrowing and fluorescence [21,22]. Although the manipulation of the microwave field amplitude and phase has been studied in a variety of systems up to date, there are no such studies of microwave pulse shaping in a multi-level atom to the best of our knowledge. The interest of shaping the microwave field is that the atomic systems are sensitive to the action of the microwave field and, hence, a slight modification in its shape will have a large impact on the transmitted probe field.

In this context, our study investigates the effect of shaping a microwave field in a V-type three-level atom using different wave forms. This configuration considers a ground state coupled to an excited state with a weak probe field. A strong laser couples the ground state to an intermediate level. A shaped microwave field is applied between the excited state and the intermediate transition. Various waveforms are analyzed, such as the tanh-hyperbolic, the Gaussian, the difference of the double Gaussian form and the PEXP (power of exponential). For sake of comparison, we also consider the conventional case in which the microwave field is considered to be a constant. Through the density matrix formalism, we derive the full analytical solutions of the diagonal and off-diagonal matrix elements. Thus, we analyze the generated coherence between the excited and the transition state after which we study the probe’s coherence under pulse shaping. Hence, in comparison with the classical scenario, where the strong field is considered to have a major role in controlling the optical properties of a multilevel atom, we show that shaping the microwave field leads to a distinct result.

This paper is structured as follows: the model is described in the first section. Next, the system dynamics is analyzed by using the density matrix formalism, followed by a discussion and the results (Section 3).

## 2. The Microwave-Forms

In electronics and digital communications, laser shaping is a technique that manipulates a transmitted waveform to achieve a desired shape in the time domain. For instance, such programmable waveforms are used to generate electromagnetic pulses with high amplitudes (HEMP), ultra-large-band pulses, and sources of gamma rays [23,24]. These waveforms can be used according to the experimental needs.

In this section, the pulses are given by various waveforms: the example of the difference of the exponential waveform is given in Figure 1a. Experimentally, this can be obtained by stretching the rising edge and compressing the falling edge, as detailed in [25].

The mathematical formula of this waveform is given by (Figure 1a)
(1)$$f\left(t\right)=k\Gamma \left(e^{-\beta^{2}t^{2}}-e^{-\alpha^{2}t^{2}}\right)$$
and the tanh-hyperbolic pulse is written as
(2)$$S\left(t\right)={\left(1-k_{2}Tanh\left(k_{3}t\right)\right)}^{2}$$
with $k_{2}$ and $k_{3}$ as arbitrary constants (see Figure 1b). The PEXP pulse is described by (Figure 1c)
(3)$$G\left(t\right)=2{\left(1-e^{-\Gamma t}\right)}^{1.5}e^{-\beta t}$$
and the Gaussian has the classical description of (Figure 1d)
(4)$$R\left(t\right)={\left(Ae^{-\alpha t}\right)}^{p}$$
with *A* is a constant and *p* is the Gaussian field order. The generation of such pulses with the mathematical description of the quotient of exponentials, rising cosine, Π pulses, sinc, power of exponentials, Gaussian pulses and modified double exponential wave-forms as well as their experimental realizations are discussed in [25].

In Figure 2, the behavior of the double Gaussian pulse shape is reported for different α and β with a fixed time t = 1.5.

Additionally, in Figure 3, the shape factor β (for the double Gaussian pulse) is a variable while α = 1.5. The pulse shape has a symmetric behavior for t varying between −1 and 1. The time t is scaled to the pulse rise time denoted by τ. The technique that relates α and β to the pulse rise time and fall edge for the double Gaussian pulse and other pulses is investigated in detail in [25].

Next, our three level atomic system is based on the classical V-type configuration (see Figure 3) in which a ground state |1〉 coupled to an upper state level |3〉 between which another intermediate state |2〉 exists [26,27,28]. Here, the Rabi frequencies coupling the transitions |i〉 to |j〉 are given by $\Omega_{ij}$, where the detunings are denoted by $\Delta_{ij}$. The probe field $\Omega_{p}$ couples the level |3〉 to |1〉, while the control field $\Omega_{c}$ drives the level |2〉 to |1〉. The transitions |3〉 and |2〉 are driven by a weak microwave field $\Omega_{s}$. The choice of this model is based on the fact that a free atom has at least two states with the same parity, between which an electrical dipole transition is not possible. This is because an electric field causes transitions to states with opposite parity. In this model, the system is described by the Hamiltonian, given by
(5)$$H=\Delta_{p}|3\rangle \langle 3|+\Delta_{c}|2\rangle \langle 2|-\left(\Omega_{c}|2\rangle \langle 1|+\Omega_{p}|3\rangle \langle 1|+\Omega_{s}|3\rangle \langle 2|\right)+cc$$

The total Hamiltonian is the sum of the atom’s proper energies and the interaction energy between the fields and the atom given by the fermionic annihilation and creation operators of the atomic system. The density matrix’s off-diagonal elements $rho_{21}$ and $rho_{23}$ oscillate at the respective driving field frequency, while the $rho_{31}$ oscillates with frequency differences between the two light fields. Under the rotating wave approximation, the density matrix formalism is used to investigate the dynamics of the system’s quantum states. Hence, the density matrix elements are the outer products of the wave function and its conjugate, where the diagonal elements describe the atomic population probabilities, and the off-diagonal ones refer to the oscillatory behavior of coherent superpositions in the system, which are the coherences. The atomic inversion is denoted by $\rho_{33}-\rho_{22}$, and we can write [29]:(6)$$\begin{array}{cc} \frac{d}{dt}\rho_{22} & =-2\gamma_{2}\rho_{22}+i\Omega_{c}\left(\rho_{12}-\rho_{21}\right)+i\Omega_{s}\left(\rho_{32}-\rho_{23}\right)\\ \frac{d}{dt}\rho_{33} & =-2\gamma_{1}\rho_{33}+i\Omega_{p}\left(\rho_{13}-\rho_{31}\right)+i\Omega_{s}\left(\rho_{23}-\rho_{32}\right) \end{array}$$

The off-diagonal elements $\rho_{ij}$ for i≠j are related to the atomic coherence in the system, and they are given by
(7)$$\begin{array}{ccc} \frac{d}{dt}\varrho_{21} & = & C1\rho_{21}+i\Omega_{c}\left(\rho_{11}-\rho_{22}\right)\\ & & -i\Omega_{p}\rho_{23}+i\Omega_{s}\rho_{31} \end{array}$$
(8)$$\begin{array}{ccc} \frac{d}{dt}\varrho_{31} & = & A1\rho_{31}+i\Omega_{p}\left(\rho_{11}-\rho_{33}\right)\\ & & -i\Omega_{c}\rho_{32}+i\Omega_{s}\rho_{21} \end{array}$$
(9)$$\begin{array}{ccc} \frac{d}{dt}\varrho_{32} & = & B1\rho_{32}+i\Omega_{p}\rho_{12}-i\Omega_{c}\rho_{31}+i\Omega_{s}\left(\rho_{22}-\rho_{33}\right) \end{array}$$
where
(10)$$\begin{array}{ccc} C1 & = & -(\gamma_{2}+i\Delta_{c})\\ A1 & = & -(\gamma_{1}+i\Delta_{p})\\ B1 & = & -(\gamma_{1}+\gamma_{2}-i\Delta_{c}+i\Delta_{p}) \end{array}$$

We recall that all atomic probabilities must sum to unity through the trace, and hence we have Trac(ρ) = 1. It is worth noting that this atomic configuration can be experimentally realized by an ensemble of $Rb^{87}$ atoms with $5S_{1/2}$, F = 1, m = 1, $5P_{1/2}$, F = 1, m = 0 and $5S_{1/2}$, F = 2, m = 1 [30] and the radiative decays are given by $\gamma_{2}=3$, $\gamma_{1}=1$.

Next, our first step is to solve the density matrix analytically by taking into account the microwave field as a constant field, where we apply the classical perturbation method (note that other pertinent methods can be used, such as the integral factor technique demonstrated in details in [5]). We derive the full solutions of all matrix elements using perturbative approximations (where the probe field is considered in first order while the other fields are in all orders). The probe coherence is given by $\rho_{31}$, while the coherence between the excited state and the intermediate level is given by $\rho_{32}$. Since the microwave field drives |3〉 to |2〉, we examine its effect on the dynamics of our system. The analytical expression of $\rho_{32}$ is
(11)$$\begin{array}{ccc} \rho_{32} & = & \frac{i\Omega p(E1+E2)}{G1+G2}-\frac{i\Omega p(E3+E4)}{G1+G2} \end{array}$$
where *E*1, *E*2, *E*3, *E*4 and *G*1 and *G*2 are given in the appendix.

Next, we numerically solve the density matrix by taking into account the microwave field as a shaped waveform. We investigate the induced variation of the probe coherence. For instance, we consider the classical case, where both the microwave and the control field are constants; we also compare the DGF effect to the Gaussian, PEXP and the tanh-hyperbolic.

## 3. Discussion

Before we investigate the probe’s coherence related to the absorption and dispersion dynamics, we first start by analyzing the generated coherence between the states |3〉 and |2〉. In Figure 4a,b, the induced coherence between the excited and the intermediate transition is reported for various microwave field amplitudes using Equation (Equation 11) (see the Appendix A for details).

Figure 4a is the imaginary of $\rho_{32}$ versus probe detuning for various values of the microwave field. We observe positive peaks (absorption) of the microwave field for negative probe detuning, and negative peaks (gain) spectrum of the unshaped microwave field for positive probe detuning. The blue and the red curves are considered, respectively, for $\Omega_{s}$ = 0.1 and $\Omega_{s}=0.4$. These two curves are almost identical. When we further increase $\Omega_{s}$, a change in the induced coherence starts to appear (see the green curves, mainly for the dispersion spectrum $Re(\rho_{32})$). In fact, the sharp and large gain peak is observed for positive probe detuning, leading to enhanced induced coherence by the unshaped microwave field through the closed-loop atomic medium. Figure 4b shows the real $\rho_{32}$ versus probe detuning for different values of the unshaped microwave field. The behavior of the dispersion spectrum of the microwave field is the same for the smaller values of $\Omega_{s}$. However, we can see an abrupt and sharp change in the anomalous dispersion spectrum of the microwave field by increasing the amplitude of $\Omega_{s}$ (see the green curve of Figure 4b). We conclude that, interestingly, when we increase the microwave field, we notice an increase in the dispersion, then a sudden change appears in the green curve. Hence, the dispersion switches from positive to negative around $\Delta_{p}=0.9\gamma$. In Figure 4c,d, we consider a microwave field of 1.26 (in units of γ), and we report the coherence for various shaped waveforms. The coherence displays distinct dynamics; the unshaped generated coherence is shown by the black curve (the highest or first curve in the graph). It is clearly seen that shaping the microwave field between the excited and the intermediate transition leads to a tunable generated coherence that would lead to a total tunable coherence for the probe’s field; see Figure 5 and Figure 6. Additionally, the generated coherence switches from positive to negative, obeying the waveform shape.

Next we analyze the total probe’s coherence. In Figure 5a, we investigate the effect of the DGF shape factor α on the dynamic of the coherence at equal detuning. We consider that 50% of the initial population is trapped in the ground state, while 20% is initially in the excited state (the other 30% is in the intermediate level). Here, $\Delta_{p}=\Delta_{c}=0.1$, $\Gamma_{1}=1$, $\Gamma_{2}=3$, $\Omega_{c}=10$, $\Omega_{p}=0.1$, $\Omega_{s}$. The DGF field is given by $f\left(t\right)=e^{-A^{2}t^{2}}-e^{-B^{2}t^{2}}$ with A = 1.5 and B = α A. *t* is in units of τ, and τ is the pulse rise time.

Figure 5b reports the real and imaginary parts of the coherence far from the detuning, respectively related to the absorption and dispersion spectra of the atomic system. The case of the unshaped (dashed) microwave field displays periodic gain and absorption prior to the steady state, where Im(ρ) = 0. For α = 0.2, 0.5, 0.7, the gain only appears below the zero-absorption line. By increasing α (see the case of α = 6), the transient gain changes from negative to positive (see the green curve in Figure 5a) before it reaches the steady state. In Figure 5b, the absorption is reduced by increasing α, whereas for a non-shaped microwave field (dashed curve), the probe is highly absorbed. In Figure 5c,d, we consider the far detuning case with $\Delta_{p}$ = 0.2, and $\Delta_{c}$ = 7. As it is shown in the figures, the red curve reports the coherence analyzed by considering both the control and the microwave field as a DGF waveform, the purple curve denotes the case where the microwave field is a Gaussian waveform and the strong pumping is a DGF field, and the black curve considers the classical scenarios where the microwave and the strong pump laser are constants. The coherence for the blue curve was calculated for a constant microwave field and a strong DGF coupling laser. Finally, the light blue curve considers a PEXP microwave pulse shape and a constant pump field, while the green curve is for a tanh-hyperbolic microwave pulse and a constant pump field. (For more details about the PEXP or the tanh-hyperbolic pulse, see [25]).

It is shown that the steady state is realized around 1.5 s for the Raman detuning, while it is reached at 2.5 s near the resonance. Moreover, the coherence is tunable by considering different shapes for the control and microwave fields, for instance, at equal detuning (near resonance), a low absorption is obtained by shaping $\Omega_{s}$ as a tanh-hyperbolic or a PEXP. We note that the lowest dispersion coefficient is seen for the far detuning case when both fields are taken as a DGF (the red curve). Finally, the highest peaks of the dispersion coefficients in the case of the green, black and blue curves Figure 6 are shifted backward by shaping the microwave field respectively with the tanh-hyperbolic, PEXP and a constant form.

Finally, in Figure 6a,b, we consider the near resonance case with $\Delta_{p}$ = $\Delta_{c}$ = 0.1, $\Gamma_{1}$ = 1, $\Gamma_{2}$ = 3, $\Omega_{p}$ = 0.7, the amplitude of the control field is $\Omega_{c}$ = 10. The shape effect is introduced through a coefficient γ that manipulates the amplitude of the microwave field. As can be shown, the absorption is controllable between t = 0 and t = 1.5 before reaching the steady state. $Re(\rho_{31})$ switches from being positive (for unshaped microwave fields) to being negative when a microwave field is shaped (with γ = 0.9). Additionally, the highest absorption is at γ = 0.9. In contrast, the absorption is reduced when γ = 0.2. In Figure 6b, changing the γ amplitude factor of the DGF micro-waveforms induces a noticeable change in the dynamics of the absorption coefficient. In addition, high absorption is displayed for the unshaped case (γ = 0.9). The transient absorptions coincide for the green (γ = 0.4), black (γ = 2.6) and the unshaped case, and the differences appear only around 0.75 s.

Finally, we observe that changing the shape of the microwave field enables the switching of the dispersion coefficient from positive to negative as well as the control of the absorption. In the far detuning case, we show that the highest peaks of the dispersion coefficients are shifted backward by shaping the microwave field with the tanh-hyperbolic, PEXP and a constant form. In contrast with the case where the control of the atomic system is achieved by varying the amplitude of the coupling field, we believe that shaping the microwave field that drives the excited state to the intermediate transition induces drastic changes in the interference pathways and hence leads to an important change in the probe’ field coherence.

## 4. Conclusions

In this work, we investigate the shape of the microwave field effect on the atomic properties of a three-level system.

Hence, in comparison with the classical scenario, where the strong field is considered to have a major role in controlling the optical properties of a multilevel atom, we show that shaping the microwave field leads to a distinct result. Hence, optical properties of the system are controllable through shaping the microwave field, while the strong laser is not altered. This might open a flexibility window in the technical use of the three level atom to better engineer atomic ensembles, such as cavities QEDs using short chains of atoms, single-mode cavities via photon transitions, chip-scale atomic clocks, etc. Hence, the microwave field can be used as an alternative to manipulate these atomic systems, which are generally controlled by the strong laser beam.

## Figures and Tables

**Figure 1 molecules-28-02096-f001:**
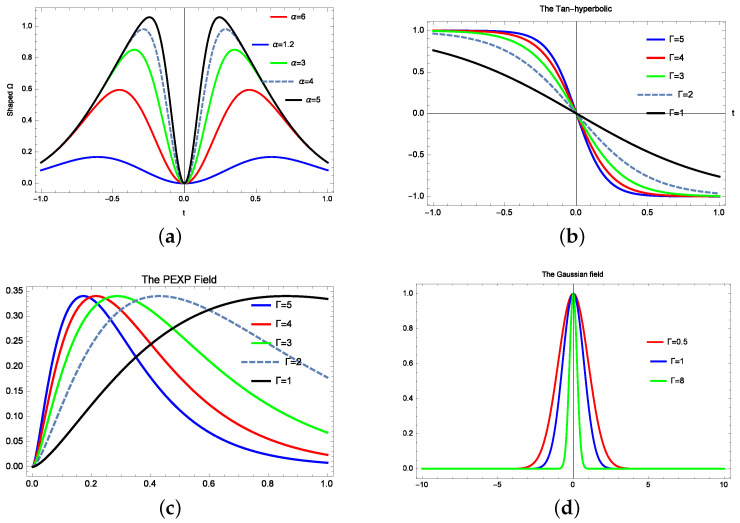
The microwave shaped field: (**a**) The double Gaussian pulse for various shape factor. (**b**) The tanh-hyperbolic pulse for various fall edge. (**c**) The power exponential pulse for various rise time. (**d**) The Gaussian pulse for various aperture factor.

**Figure 2 molecules-28-02096-f002:**
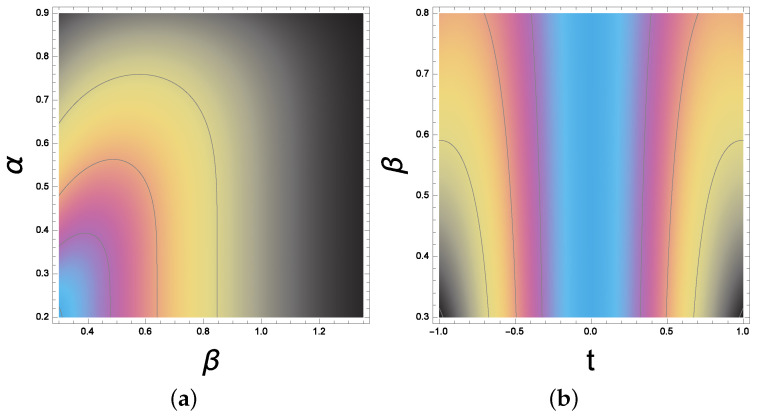
The pulse dynamics: the double exponential waveform for various shape factors. The pulse dynamics: (**a**) the double exponential waveform for various shape factors. β is a variable while α = 1.5. (**b**) The pulse shape has a symmetric behavior for t varying between −1 and 1 which is interpreted by the colors around the blue centered line.

**Figure 3 molecules-28-02096-f003:**
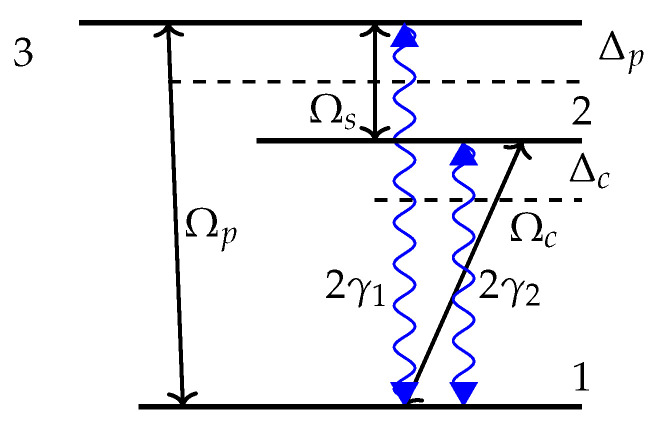
A classical configuration of a V-type three-level atom where $\Omega_{s}$ is a shaped microwave field.

**Figure 4 molecules-28-02096-f004:**
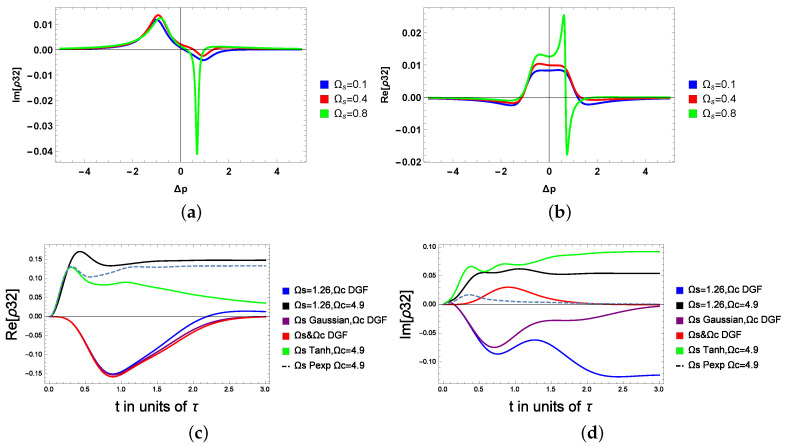
The generated coherence between the excited and the intermediate state: (**a**,**b**) for variable amplitude of the microwave field, (**c**,**d**) for shaped microwave field.

**Figure 5 molecules-28-02096-f005:**
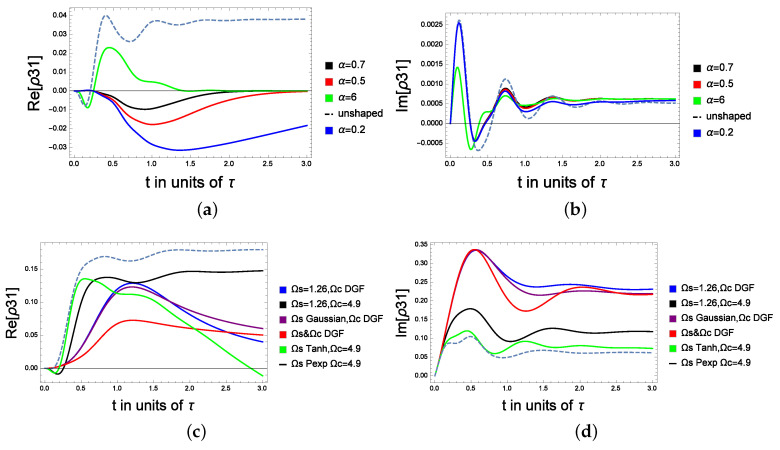
The probe coherence between the excited and the ground state under the effect of the microwave shaped coherence in the excited and the intermediate state: (**a**) The real part of the coherence for various shape factors α, $\Omega_{p}$ = 0.1, $\Delta_{p}$ = $\Delta_{c}$ = 0.1. (**b**) The imaginary part of the coherence for various shape factors α, $\Omega_{p}$ = 0.1, $\Delta_{p}$ = $\Delta_{c}$ = 0.1. (**c**) The real part of the coherence: $\Delta_{p}$ = 0.2 and $\Delta_{c}$ = 7. (**d**) The imaginary part of the coherence: $\Delta_{p}$ = 0.2 and $\Delta_{c}$ = 7.

**Figure 6 molecules-28-02096-f006:**
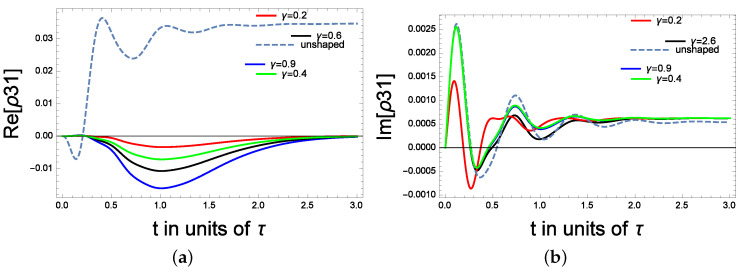
(**a**) The coherence for tunable amplitude of the microwave field: the real part. (**b**) The coherence for tunable amplitude of the microwave field: the imaginary part.

## Data Availability

All data are available upon reasonable request from the corresponding author.

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
