# Peer review of "Shaped Microwave Field in a Three-Level Closed Loop Dense Atomic System"

_molecules, 2023, doi:10.3390/molecules28052096_

Round 1
Author Response
February 14/02/2023
Letter to the Editor on Manuscript molecules-2208237  
Dear Editor,
Thank you for reviewing our manuscript entitled ‘’ Shaped Microwave Field in a Three-Level Closed Loop Dense Atomic System.’’
We do highly appreciate Reviewer's insightful comments on improving our manuscript and we greatly acknowledge the time and effort of both reviewer 1 and reviewer 2.
Our answers are listed below, following the reviewers’ comments.
Sincerely,
Nadia Boutabba
The authors report on a theoretical work that investigates the control of the optical properties of a 3-level atom that is coupled to two laser fields, one strong and one weak, in addition to a microwave field coupling the two excited states. In their study they show that the optical properties of the system can be controlled especially when using the shaping the microwave field. Their results may be applied to realistic atomic states, such as in Rb as they refer in the manuscript, although I am not aware of any attempts towards this direction. Being not a guru of the field, I find their work quite interesting that deserves publication. The manuscript suffers from some English and typo errors, although it is well written and quite clearly presented, while the authors should have also paid more attention to the presentation of their figures as I suggest below. The manuscript can be published after considering the following:
Line 17. OCT does not seem to correspond to “coherent control of light”.
We thank the reviewer for drawing our attention to this typos error. OCT is the optical coherence tomography which is a branch of the optical coherent control of light, that is applied to the photochemical reactions. Hence, we corrected line 17 from:
‘’For instance, coherent control of light
(OCT) has made significant contributions to our understanding of the manipulation of 17
photo-chemical reactions’’
To be now:
(Answer)
For instance, coherent control of light
has made significant contributions to our understanding of the manipulation of atomic systems such as the photo-chemical reactions based on the optical coherence tomography (OCT).
Line 35. “Known” should read “known”.
We corrected the error.
Line 52. The acronym “EIT” is not defined.
We corrected to be Electromagnetically induced transparency
Lines 59 & 75. “Multi-level” should read “multi-level”.
We corrected this error and we verified that it is corrected in all the text.
Figure 1. The graphs should be aligned. The legends inside the graphs are too small to read. In the caption correct the sentences to plural. Also use (a)..(d) inside the graphs for better correspondence with the caption.
We corrected the mistakes and adjusted the space between the figures
Figure 2. There is no color scale. A better explanation of the figures should be goven at the caption.
We added the clarifications in the caption.
\caption{The pulse dynamics: the double exponential waveform for various shape factors. $\beta$ is a variable while $\alpha$=1.5. The pulse shape has a symmetric behavior for t varying between -1 and 1 which is interpreted by the colors around the blue centered line}
Figure 3. Explain all the symbols that are in the figure to the caption of the figure for completeness.
We added an explanation in the caption
Line 91. “. while” should read “, while”.
Thank you we amended the manuscript.
Line 101. “Fig. 3” should be “Fig. 2”. Thank you, we corrected. This now line 102.
Line 120. All the “rho”s should be given with the mathematical rho.
We re-read all the manuscript and corrected the error.
Line 146. “Where” should read “where.
We corrected this error
Figure 4. Also use (a)..(d) inside the graphs for better correspondence with the caption. Perhaps justifying Imaginary and Real graphs under the same column would be better (as in Fig. 5).
We amended the graphs as required.
Figure 5. Also use (a)..(d) inside the graphs for better correspondence with the caption.
We corrected this in the graph as advised. Thank you very much, our manuscript is now more clear and greatly improved.
Answers to reviwer2:
Review of the manuscript with Manuscript ID: molecules-2208237
In this work the authors have investigated the atomic properties of a three-level system
under the effect of a shaped microwave field. The system under study is simultaneously
driven by a powerful laser pulse and a weak constant probe that drives the ground state
to an upper level for seek of comparison. The authors investigate the tan-hyperbolic, the
Gaussian and the power of exponential microwave-form in the system.
The results reveal that the shape of the external microwave
field has a significant impact on the absorption and dispersion coefficient dynamics.
Comparison with classical description, where usually the strong pump laser is considered
to have a major role in controlling the absorption spectrum, shows that shaping the
microwave field lead to distinct results the microwave field can be used as an alternative
to manipulate these atomic systems, which are generally
controlled by the strong laser beam.
The mathematical description of the applied model is given in details and the generated
pulses are discussed.
I find this work interesting and it could be of use to compute optical properties of systems
in a controllable manner through shaping the microwave field while the strong laser is
not altered.
Question 1
1) I found some problems along the text which should be improved.
Answer1:
We edited our manuscript carefully, and we checked all the graphs.
Question2:
2) My main concern about this work is that it is important for the authors to show the
application of this method to some systems of interest with experimental
comparison before the manuscript is considered for its publication in Molecules
Answer 2:
We would like to thank you the reviewer for this scientific question. We updated our manuscript in line .
In fact, the experimental study for the transition probability of two level atoms driven by shaped pulse is presented in the references below. The various shaped pulse s i.e., Lorentzian, hyperbolic secant, Gaussian, Lorentzian squared, and hyperbolic secant squared shaped pulse has been used to study the temporal dynamics of the two level atom. It has been shown that the population transfer and spectral properties of the medium dramatically changes by considering the temporal shaped pulse [31]. The microwave shaped pulse has shown to control the ground state coherence experimentally of the closed three level atomic medium. The two photon resonance condition generated the coherence while the three photon resonance amplified the probe light significantly with 98.8 %. visibility [32]. Recent experimental and theoretical study of the interaction of shapes i.e., Gaussian, rectangular, exponential, hyperbolic-secant, and squared hyperbolic-secant with qubit showed its significant dependence on the transition probability profile [33]. Higher fidelity quantum logic gate has been achieved using Gaussian driving field, which may be helpful quantum simulation and quantum computation in the neutral-atomic system [34]. The pulse shaping systems developed and can control many molecular and atomic processes. The experimental realization of the shaped optical pulse revolutionize the communication process i.e, terabit/sec data communications and quantum computing and.
[31]. C. W. S. Conover “Effects of pulse shape on strongly driven two-level systems” Phy. Rev. A. 84 063416, 2011.
[32]. K. V. Adwaith, K. N. pradosh, J. K. Saaswath, F. Bretenaker, and A. Narayanan,” Microwave controlled ground state coherence in an atom-based optical amplifier” OSA Continuum 4, 2021.
[33] arXiv:2201.05994v1 [quant-ph] 16 Jan 2022
Single Gaussian temporal pulse modulated controlled-Z gate of neutral atoms under symmetrically
optical pumping
X. X. Li,1 X. Q. Shao,1, 2, ∗ and Weibin Li3, †
[34] Optical pulse shaping approaches to coherent control
Debabrata Goswami∗
Tata Institute of Fundamental Research, Homi Bhabha Road, Mumbai 400 005, India
Accepted 1 September 2002
editor: J. Eichler

Reviewer 2 Report
Dear Editor,
The authors report on a theoretical work that investigates the control of the optical properties of a 3-level atom that is coupled to two laser fields, one strong and one weak, in addition to a microwave field coupling the two excited states. In their study they show that the optical properties of the system can be controlled especially when using the shaping the microwave field. Their results may be applied to realistic atomic states, such as in Rb as they refer in the manuscript, although I am not aware of any attempts towards this direction. Being not a guru of the field, I find their work quite interesting that deserves publication. The manuscript suffers from some English and typo errors, although it is well written and quite clearly presented, while the authors should have also paid more attention to the presentation of their figures as I suggest below. The manuscript can be published after considering the following:
Line 17. OCT does not seem to correspond to “coherent control of light”.
Line 35. “Known” should read “known”.
Line 52. The acronym “EIT” is not defined.
Lines 59 & 75. “Multi-level” should read “multi-level”.
Figure 1. The graphs should be aligned. The legends inside the graphs are too small to read. In the caption correct the sentences to plural. Also use (a)..(d) inside the graphs for better correspondence with the caption.
Figure 2. There is no color scale. A better explanation of the figures should be goven at the caption.
Figure 3. Explain all the symbols that are in the figure to the caption of the figure for completeness.
Line 91. “. while” should read “, while”.
Line 101. “Fig. 3” should be “Fig. 2”.
Line 120. All the “rho”s should be given with the mathematical rho.
Line 146. “Where” should read “where.
Figure 4. Also use (a)..(d) inside the graphs for better correspondence with the caption. Perhaps justifying Imaginary and Real graphs under the same column would be better (as in Fig. 5).
Figure 5. Also use (a)..(d) inside the graphs for better correspondence with the caption.
Author Response

(The authors gave the same response as above.)

Round 2
Reviewer 1 Report
Report of the revised version of the manuscript with ID molecules-2208237
The authors have addressed all points made by the reviewer in a satisfying form.
In my opinion the present version of the manuscript can be now accepted for its publication.